# GIS-Based Approach Applied to Study of Seasonal Rainfall Influence over Flood Vulnerability

Rita de Cássia Freire Carvalho [1], Taís Rizzo Moreira [1], Kaíse Barbosa de Souza [1], Gizely Azevedo Costa [1], Sidney Sara Zanetti [1], Kargean Vianna Barbosa [2], Cláudio Barberini Camargo Filho [1], Maiara Rodrigues Miranda [2], Plinio Antonio Guerra Filho [3], Aline Ramalho dos Santos [1], Antonio Henrique Cordeiro Ramalho [1], Elias Secretário Armando Ferreira [1], Emanuel França Araújo [1], Felipe Patricio das Neves [1], Jeneska Florencio Vicente de Lima [2], Julia Siqueira Moreau [1], Leonardo Leoni Belan [2], Marcelo Otone Aguiar [1], Rodrigo Gomes Gorsani [4], Simony Marques da Silva Gandine [5] and Alexandre Rosa dos Santos [1,*]

1    Postgraduate Programme in Forest Sciences, Espírito Santo Federal University (UFES), 316 Governador Lindemberg Avenue, Jerônimo Monteiro 29550-000, Brazil
2    Postgraduate Programme in Agronomy, Espírito Santo Federal University (UFES), Alto Universitário, s/n, Alegre 29500-000, Brazil
3    Chapadinha Sciences Center, Maranhão Federal University (UFMA), Highway BR222, km04, Chapadinha 65500-000, Brazil
4    Postgraduate Programme in Botany, Biological Sciences Center 2, Plant Biology Department, Viçosa Federal University, University Campus, Viçosa 36570-900, Brazil
5    Postgraduate Programme in Agrochemistry, Espírito Santo Federal University (UFES), Alto Universitário, s/n, Alegre 29500-000, Brazil
*    Correspondence: mundogeomatica@yahoo.com.br; Tel.: +55-28-999260262

**Abstract:** Flooding occurrence is one of the most common phenomena that impact urban areas, and this intensifies during heavy rainfall periods. Knowing the areas with the greatest vulnerability is of paramount importance as it allows mitigating actions to be implemented in order to minimize the generated impacts. In this context, this study aimed to use Geographic Information System (GIS) tools to identify the areas with greater flooding vulnerability in Espírito Santo state, Brazil. The study was based on the following methodological steps: (1) a Digital Elevation Model (DEM) acquisition and watersheds delimitation; (2) maximum and accumulated rainfall intensity calculations for the three studied periods using meteorological data; (3) a land use and occupation map reclassification regarding flood vulnerability and fuzzy logic application; (4) an application of Euclidean distance and fuzzy logic in hydrography and water mass vector variables; (5) a flood vulnerability model generation. Based on the found results, it was observed that the metropolitan and coastal regions presented as greater flood vulnerability areas during the dry season, as in these regions, almost all of the 9.18% of the state's area was classified as highly vulnerable, while during rainy season, the most vulnerable areas were concentrated in Caparaó and in the coastal and immigration and metropolitan regions, as in these regions, almost all of the 12.72% of the state's area was classified as highly vulnerable. In general, by annually distributing the rainfall rates, a greater flood vulnerability was observed in the metropolitan and coastal and immigration regions, as in these areas, almost all of the 7.72% of the state's area was classified as highly vulnerable. According to the study, Espírito Santo state was mostly classified as a low (29.15%) and medium (28.06%) flood vulnerability area considering the annual period, while its metropolitan region has a very high flood vulnerability risk. Finally, GIS modeling is important to assist in decision making regarding public management and the employed methodology presents worldwide application potential.

**Keywords:** floods; geotechnologies; urban flooding; rainfall; land use and occupation; fuzzy logic

## 1. Introduction

Flooding can be characterized by the rising or overflowing of a water flow. It is considered a natural and severe phenomenon, such as droughts and hurricanes, and it is influenced by regional characteristics such as soil and vegetation types and weather conditions [1]. This phenomenon is considered a "natural disaster" since it occurs in areas where human presence predominates, thereby causing great devastation, economic impacts and lives being loss [2,3].

Flooding usually occurs after heavy rainfall and causes damage mainly in urban locations where there is disorderly occupation as well hazardous areas. Cities are proximal to rivers due to the human physiological need for water resources [4].

Since the earliest civilizations, humans sought to build their cities near rivers in search of water for irrigation, animal desedentation and fertile land. Thus, urban areas face the flood risk challenge not only due to climate change, but also as an urbanization process consequence [5].

A flood is an extreme weather event [6], which puts at risk the population that lives mainly in the watercourses vicinity and areas with fewer slopes. In recent years, there have been flooding reports in countries within the Indu River basin, such as in 2010 in Pakistan, in the mountainous region of Brazil in January 2011 which incurred landslides, and in March 2019, in the Indonesian province of Papua [2]. In addition, it is known that both natural and anthropogenic processes impact the hydrological dynamics of floodplains, rivers and coastal regions, as it alters the surface runoff and infiltration of water [7].

In Brazil in 2013, Espírito Santo and Minas Gerais states were affected by the strongest rainfall event of the last 90 years. According to meteorologists, the Espírito Santo rainfalls have cost the public coffers around \$260 million, caused 30 deaths and left around 50,000 people homeless [8]. Thus, it is important to study and identify the main flood risk areas in the state in order to provide mitigation and preventive measures against major floods.

It is of fundamental importance to implement methodologies to prevent the floods occurring in order to mitigate the losses that are caused by them. In this sense, there is a wide variety of techniques that seek to generate a flood vulnerability model, and among them are the use of geotechnologies, with the aid of fuzzy logic, and the Analytic Hierarchy Process (AHP). Both methodologies enable the study of flood vulnerability. Several factors influence flood vulnerability, such as relief characteristics, the watershed compactness coefficient, rainfall intensity and distribution, land use and occupation and soil types. A Geographic Information Systems (GIS)-based approach makes it possible to identify the most susceptible areas to flooding, thereby assisting in decision-making that is aimed at mitigating the flooding impacts [9,10].

Fuzzy logic is used to standardize the criteria on a scale from 0 to 1, with values that are close to 0 being considered the least favorable ones and values that are close to 1 being the most favorable ones [11–13]. The AHP method, in turn, measures each variable's importance using hierarchy levels through a pairwise comparison, and thus provides support for the decision making [14].

Some developed research studies demonstrate the importance and applicability of the geotechnology use combining Fuzzy logic and the AHP method. Some studies can be highlighted such as those that were developed by the authors of: [15], who evaluated the risk of flooding in the subway system using the original AHP method and the AHP method based on triangular fuzzy numbers; [16], who used the AHP and AHP-Fuzzy method to map the potential groundwater recharge zone in the arid areas of Ewaso Ng'iro–Lagh Dera River Basin, Kenya; [17], who elaborated a flood susceptibility mapping of the West Ghat coastal belt using the AHP method.

Flood risk mapping is an important and widely used tool for disaster alert and prevention [3]. Therefore, the prediction of flooding scenarios in urbanized areas and their understanding are of great importance to assist in decision making that is related to flood vulnerability factors, which may contribute to urban planning and avoid the main consequences

of these disasters [18]. It is possible to create measures that reduce flooding vulnerability and promote the infiltration and storage of water through urban planning, such as the implementation of areas with green coverage and rainfall water reuse technology [19].

Climate change is strongly associated with increased flood vulnerability, especially in coastal cities, due to the possibility of sea level rises occurring [20]. Those rises, associated with growing urbanization and extreme rainfall events occurrence, puts the populations in coastal and riverside areas at risk, which generates significant social, financial and economic impacts [21]. Given this scenario and the need to work with a more robust methodology for flood vulnerability mapping, this study aimed to apply a GIS-based approach, combining fuzzy logic and the AHP method, to study the seasonal rainfall influence on flood vulnerability in Espírito Santo state.

## 2. Materials and Methods

### 2.1. Study Area

#### 2.1.1. Climatic Characterization

The study area comprises Espírito Santo state, Southeastern Brazil, which has a total area of 46,052.64 km$^2$, and is located between parallels 17°53′29″ and 21°18′03″ south latitude and meridians 39°41′18″ and 41°52′45″ longitude west of Greenwich, bordering the Atlantic Ocean to the east, Bahia state to the north, Minas Gerais state to the west, and Rio de Janeiro state to the south (Figure 1).

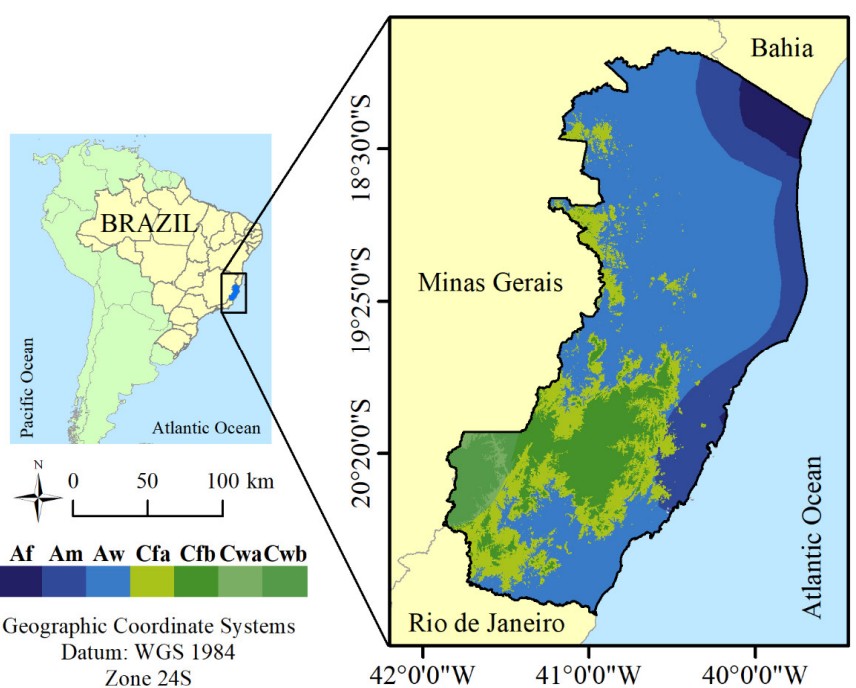

**Figure 1.** Study area geographic location containing the Köppen climate classification.

According to Köppen climate classification, the state has seven climate types: Af, humid tropical climate; Am: humid or humid tropical climate; Aw: humid tropical climate with dry winter; Cfa: humid temperate climate with hot summer; Cfb: humid temperate climate with temperate summer; Cwa: humid temperate climate with dry winter and hot summer; Cwb: humid temperate climate with dry winter and temperate summer; Cwc: humid temperate climate with dry winter and short, cool summer.

Rainfall in Espírito Santo is concentrated in the summer, and the state has an average maximum rainfall of approximately 40.2 mm (Figure S1, Supplementary Materials) throughout the year, reaching 31.3 mm during dry periods and 51.6 mm during rainy periods. The annual accumulated rainfall is approximately 1453.1 mm, reaching 477.4 mm during dry periods and 1014.7 mm during rainy periods (Figure S1, Supplementary Materials).

### 2.1.2. Geomorphological Characterization

Most of the state's area is characterized as a plateau, with altitudes ranging from 0 to 2862.5 m (Figure S2, Supplementary Materials). It also has a rugged relief, with slope values ranging from 0 to 75% (Figure S3, Supplementary Materials). The predominant soil types are: Oxisols (48.6%), Ultisols (26.0%), Inceptisols (12.1%), Entisols (4.2%), Gleysols (2.8%), Nitisols (2.3%), Phaeozem (0.9%), Spodosols (0.8%) and Histosols (0.1%) (Figure S4, Supplementary Materials).

The state is included in its entirety in the Atlantic Forest biome, an important Brazilian biome, which is distributed throughout the country as a continuous strip along the Atlantic coastal plain [22]. Despite the state being part of this considerable biome, only 15.18% of the original native vegetation of the Atlantic Forest is intact (Figure S4, Supplementary Materials). The predominant land use and occupation class in the state is pasture, occupying approximately 41.97% of the area.

### 2.1.3. Socioeconomic Characterization

According to the 2010 Brazilian demographic census, Espírito Santo's total population was 3,514,952 inhabitants, and its average demographic density was 76.25 people/km$^2$ (Figure S6, Supplementary Materials) [23]. The municipalities with the highest demographic densities are Vitória, Vila Velha, Serra, Viana, Cariacica and Guarapari, which are recognized by the state government as the Metropolitan Region.

The state has other regions that deserve to be highlighted, namely: Capixaba Mountains Region; Immigrants Region; Caparaó Region; Green and Water Region; Doce Pontões Capixabas Region; Coffee and Valleys Region; Doce Terra Morena Region; Stones, Bread and Honey Region; Coastal and Immigration Region, the latter comprising the municipalities of Alfredo Chaves, Anchieta, Iconha, Itapemirim, Marataízes and Presidente Kennedy [24].

The main economic activities that have been developed in the state are agriculture, livestock and mining [25], with there being an emphasis on the production of coffee, banana, sugar cane and coconut in agriculture (Figure S5, Supplementary Materials).

### 2.2. *Methodological Steps*

Step 1—Relief

Digital elevation model was obtained using Shuttle Radar Topographic Mission (SRTM) data with a spatial resolution of 30 m, and it was made available by National Institute for Space Research (INPE) (www.inpe.br (accessed on 15 September 2021)). Through the DEM, state watersheds were delimited and the watershed compactness coefficient (*Kc*) was obtained.

$$Kc = 0.28 \times \frac{P}{\sqrt{A}} \tag{1}$$

where *Kc* is the compactness coefficient (dimensionless), *P* is the perimeter (km) and *A* is the watershed area (km$^2$). This coefficient is a dimensionless number that varies with basin shape, regardless of its size. Thus, the more irregular it is, then the greater compactness coefficient is [12].

Step 2—Meteorological data

Rainfall data were acquired from 21 automatic stations in Espírito Santo state and its neighboring regions, and they were made available by the Meteorology National Institute (INMET) (portal.inmet.gov.br (accessed on 25 September 2021)). Use of data from neighboring regions was justified to enable us to perform statistical interpolations. Obtained data from seasons were used to calculate annual average maximum rainfall, dry and rainy periods average maximum rainfall, annual accumulated rainfall, dry and rainy periods accumulated rainfall values. After calculating average maximum rainfall and accumulated rainfall values for both periods, the station data were integrated into a point vector file. Each point was spatialized according to its geographical coordinates in the Transverse Mercator Projection (UTM) using ellipsoid WGS 84.

After average maximum rainfall and accumulated rainfall data spatialization, files were interpolated, reclassified, and at end of this step, fuzzy logic was applied.

Step 3—Land Use and Occupation and Soil Classes

The vector files of land use and occupation and soil classes were obtained from the Integrated System of Geospatial Bases of the State of Espírito Santo (Geobases) (geobases. s3.es.gov.br/minio/public/ (accessed on 22 September 2021)).

The land use and occupation map was obtained from the interpretative analysis (photointerpretation) and the manual vectorization of the limits between the land use and the land cover classes of the orthophotomosaics. The orthophotomosaic was generated from the multispectral aerophotogrammetric survey covering the entire state with a spatial resolution of 25 cm [26].

Espírito Santo's soil recognition map was based on the soil map that was published by Projeto Radam Brasil, which is available on the IBGE website (www.ibge.gov.br (accessed on 20 September 2021)), and it has been updated based on soil mapping units [27]. The maps were reclassified according to the flood vulnerability, and then, fuzzy logic was applied.

Step 4—Hydrography

The vector files of hydrography and water masses were obtained from Geobases (geobases.s3.es.gov.br/minio/public/ (accessed on 22 September 2021)). The entire state hydrography mapping was carried out using stereoscopic restitution processes on oriented models with the objective of extracting, from the aerophotogrammetric survey (25 cm spatial resolution), specific features such as watercourses, lakes and ponds, among others.

Watercourses such as rivers, streams and channels were restored by stereocompilation processes to be polylines, aiming to build a structure of a single-line network that is topologically consistent in terms of connectivity with a defined orientation from source to mouth in a graph (tree) format and without discontinuities or double confluences. Water masses such as lakes, ponds, dams and rivers or canals were rendered by stereocompilation processes to be polygons [26]. The Euclidean distance procedure has been applied to the hydrography image following the logic whereby, the farther away the area is from the hydrography, then the lower the flood vulnerability will be, and this was one of the image processing steps.

Step 5—Flood Vulnerability Model

Variables were grouped into two large groups as follows: Group 1 (the basin compactness coefficient, slope, digital elevation model, annual average maximum rainfall, dry period average maximum rainfall, rainy season average maximum rainfall, annual accumulated rainfall, dry season accumulated rainfall, rainy season accumulated rainfall and proximity to hydrographies elements); Group 2 (the land use and occupation, soil classes and proximity to water masses elements). Then, we used the Analytic Hierarchy Process (AHP) that was developed by [28] which proposes the solving of the problem by hierarchical levels, which generated coefficients that were used a posteriori to calculate the flood vulnerability model [29].

Flood vulnerability model was calculated using field calculator function that associated the coefficients that were generated by AHP method with fuzzy logic variables in their respective groupings, thus generating three vulnerabilities: two of them were for Group 01 (*Vul*1), corresponding to dry and rainy periods, and one of them was for Group 02 (*Vul*2) (Equations (2) and (3), respectively).

$$Vul1 = \beta 1 \times V1 + \beta 2 \times V2 + \beta 3 \times V3 + \beta 4 \times V4 + \beta 5 \times V5 + \beta 6 \times V6 + \beta 7 \times V7 + \beta 8 \times V8 + \beta 9 \times V9 + \beta 12 \times V12 \tag{2}$$

$$Vul2 = \beta 10 \times V10 + \beta 11 \times V11 + \beta 13 \times V13 \tag{3}$$

where *V*1 is the basin compactness coefficient, *V*2 is the slope, *V*3 is the Digital Elevation Model (DEM), *V*4 is the annual average maximum rainfall, *V*5 is the dry period average maximum rainfall, *V*6 is the rainy season average maximum rainfall, *V*7 is the annual

accumulated rainfall, *V*8 is the dry period accumulated rainfall, *V*9 is the rainy period accumulated rainfall, *V*10 is the land use and occupation, *V*11 is the soil classes, *V*12 is the proximity to hydrographies, *V*13 is the proximity to water masses and *β*1 to *β*13 are the coefficients that were obtained for each variable after applying the AHP method.

Then, three vulnerability maps were obtained through fuzzy logic use, associating the Group 01 maps with the Group 02 map. In the end, the three maps were vectorized.

The methodological flowchart containing the necessary steps for the flood vulnerability model development for Espírito Santo state is presented in Figure 2.

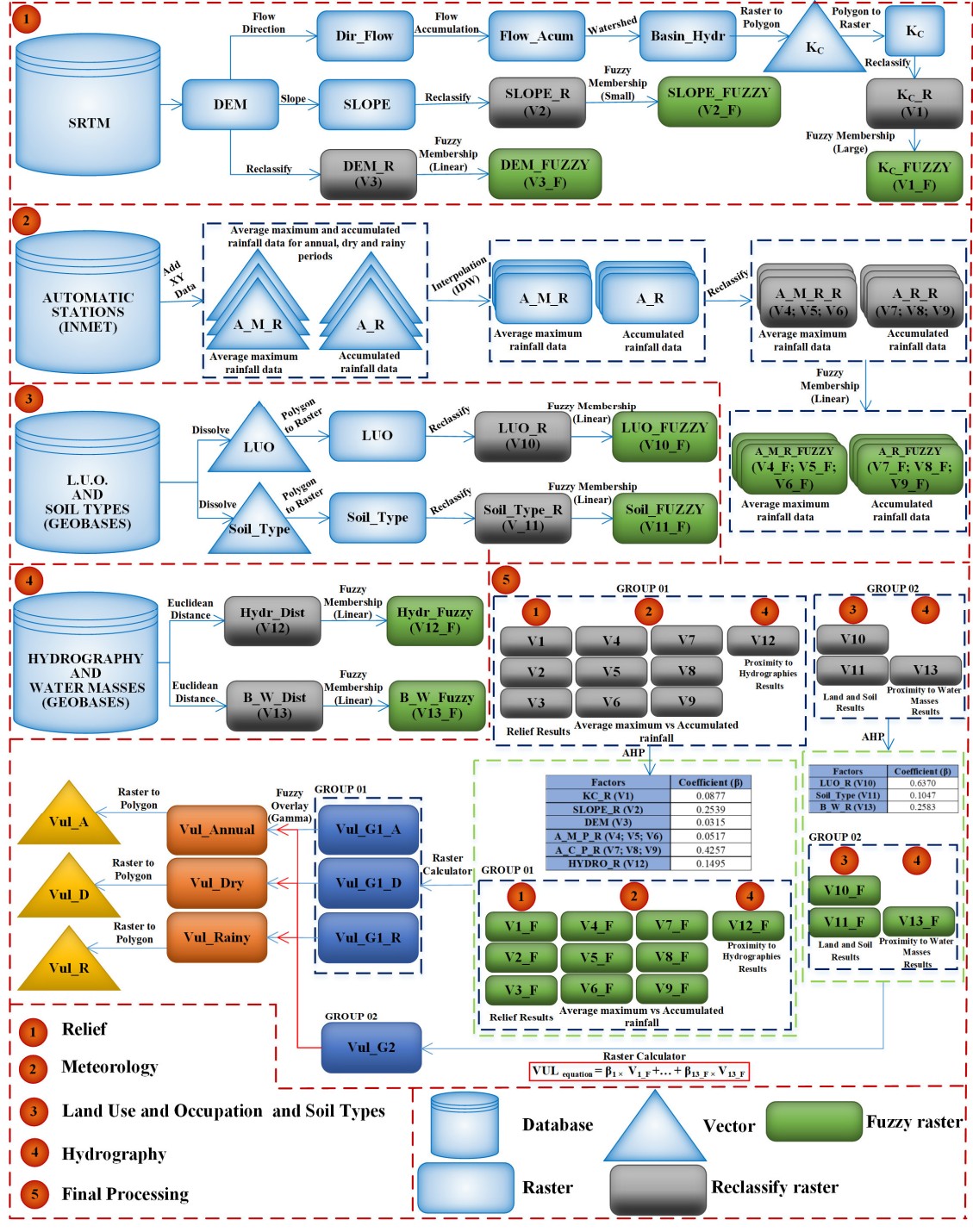

**Figure 2.** Methodological flowchart containing the necessary steps for the flood vulnerability model development for Espírito Santo state.

## 3. Results

Fuzzy logic is a form of multivariate logic that is used to standardize the criteria on a scale ranging from 0 to 1 [13]. It allows, in turn, one to analyze data that present some kind of inaccuracy and are characterized by a membership function in which each variable of interest is associated with a pertinence value [30]. In fuzzy logic, 0 value represents a falsehood and 1 is considered to be an absolute truth [31].

Regarding flooding, the closer to 0 that the value is, then the lower the flood vulnerability is, while the closer that the value is to 1, it means the opposite. Adjusted membership functions for each studied variable are presented in Figures 3 and 4.

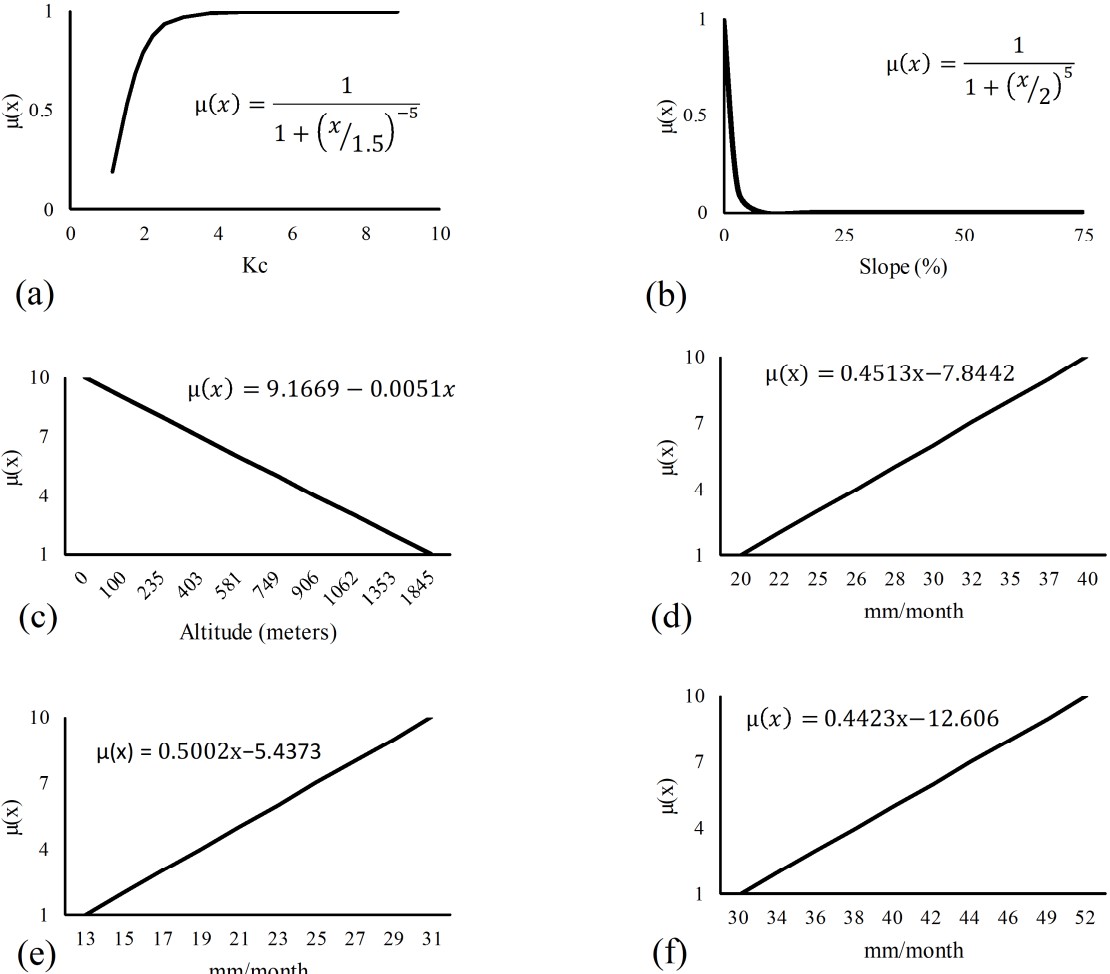

**Figure 3.** Membership functions applied to the flood vulnerability model variables. (**a**) Compactness coefficient; (**b**) slope; (**c**) Digital Elevation Model (DEM); (**d**) annual average maximum rainfall; (**e**) dry period average maximum rainfall; (**f**) rainy period average maximum rainfall.

The annual average maximum rainfall, dry period average maximum rainfall, rainy period average maximum rainfall, annual accumulated rainfall, dry period accumulated rainfall and rainy period accumulated rainfall variables were adjusted by increasing the linear function. DEM, the proximity to hydrographies and the proximity to water masses variables were adjusted by decreasing the linear function. The slope variable was adjusted by decreasing the sigmoid function (small), and the basin compactness coefficient showed increasing sigmoidal behavior (large).

Representative histograms of pixel frequency percentage in each fuzzy class (0.0–0.25; 0.25–0.5; 0.5–0.75; 0.75–1.0) are shown in Figures 5 and 6.

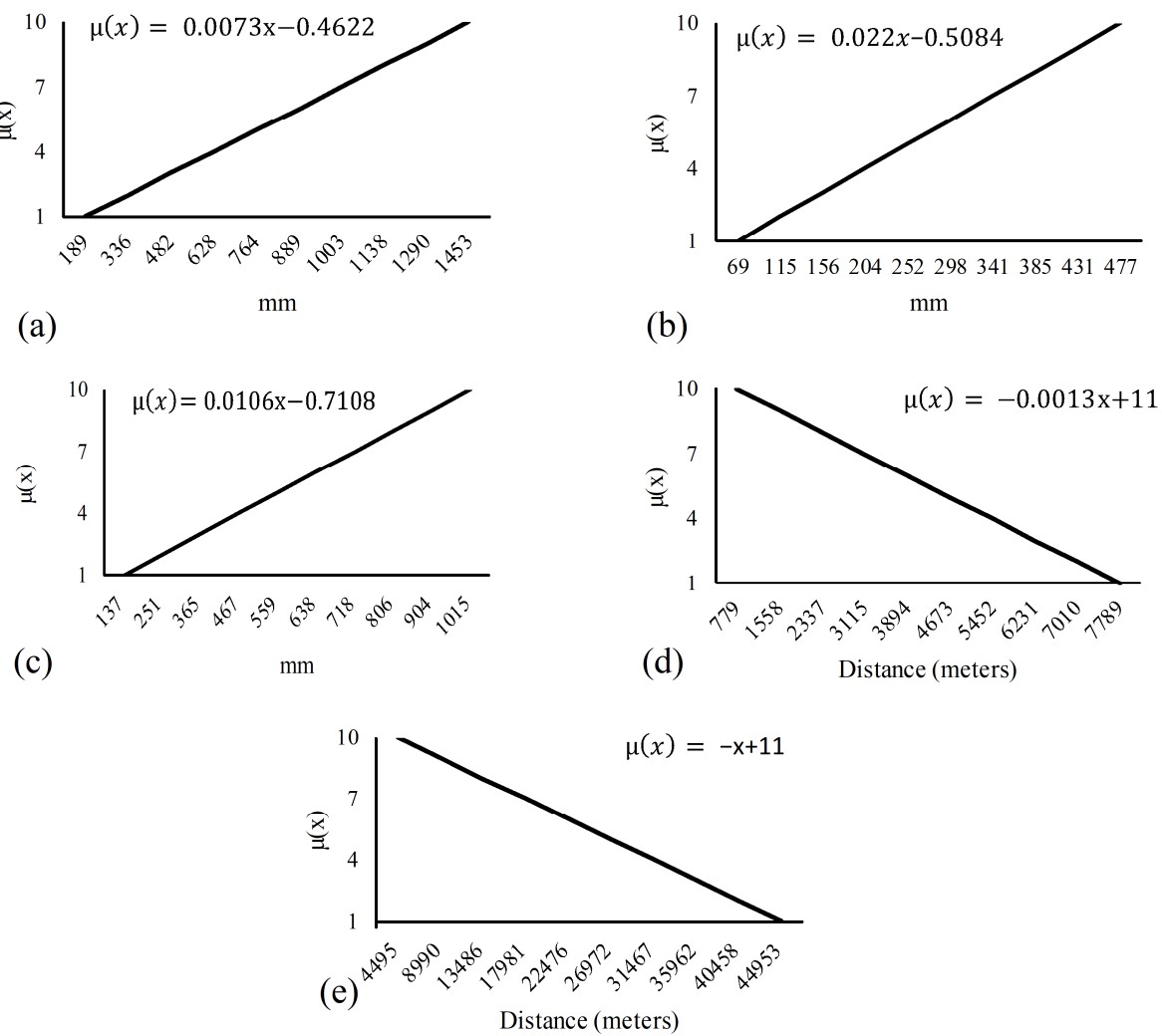

**Figure 4.** Membership functions adjusted for each studied variable of the flood vulnerability model. (**a**) Annual accumulated rainfall; (**b**) dry period accumulated rainfall; (**c**) rainy period accumulated rainfall; (**d**) proximity to hydrographies; (**e**) proximity to water masses.

According to the Group 01 pixels frequency results, the slope variable presented the highest pixels percentage (98.3%) in the less prone to vulnerability fuzzy class (0.0 to 0.25), which was followed by the dry period average maximum rainfall variable (45.1%). The other variables presented values that are below 29.9% for the less prone to vulnerability fuzzy class.

The compactness coefficient and proximity to hydrographies variables presented the highest equivalent percentage values, 99.9% and 98.1%, respectively, for the most prone to vulnerability fuzzy class (0.75 to 1), which were followed by the Digital Elevation Model variable (70.7%). The other variables presented values lower than 27.8% for the same fuzzy class.

Regarding the pixels frequency for Group 2, the soil classes variable (Figure 4) presented the highest percentage (52.9%) for the less prone to vulnerability fuzzy class, while the proximity to water masses variable presented the highest percentage for the most prone to vulnerability fuzzy class, corresponding to 76.2%. The land use and occupation variable presented the highest percentage value for the 0.25 to 0.5 fuzzy class, which was equivalent to 55.3%.

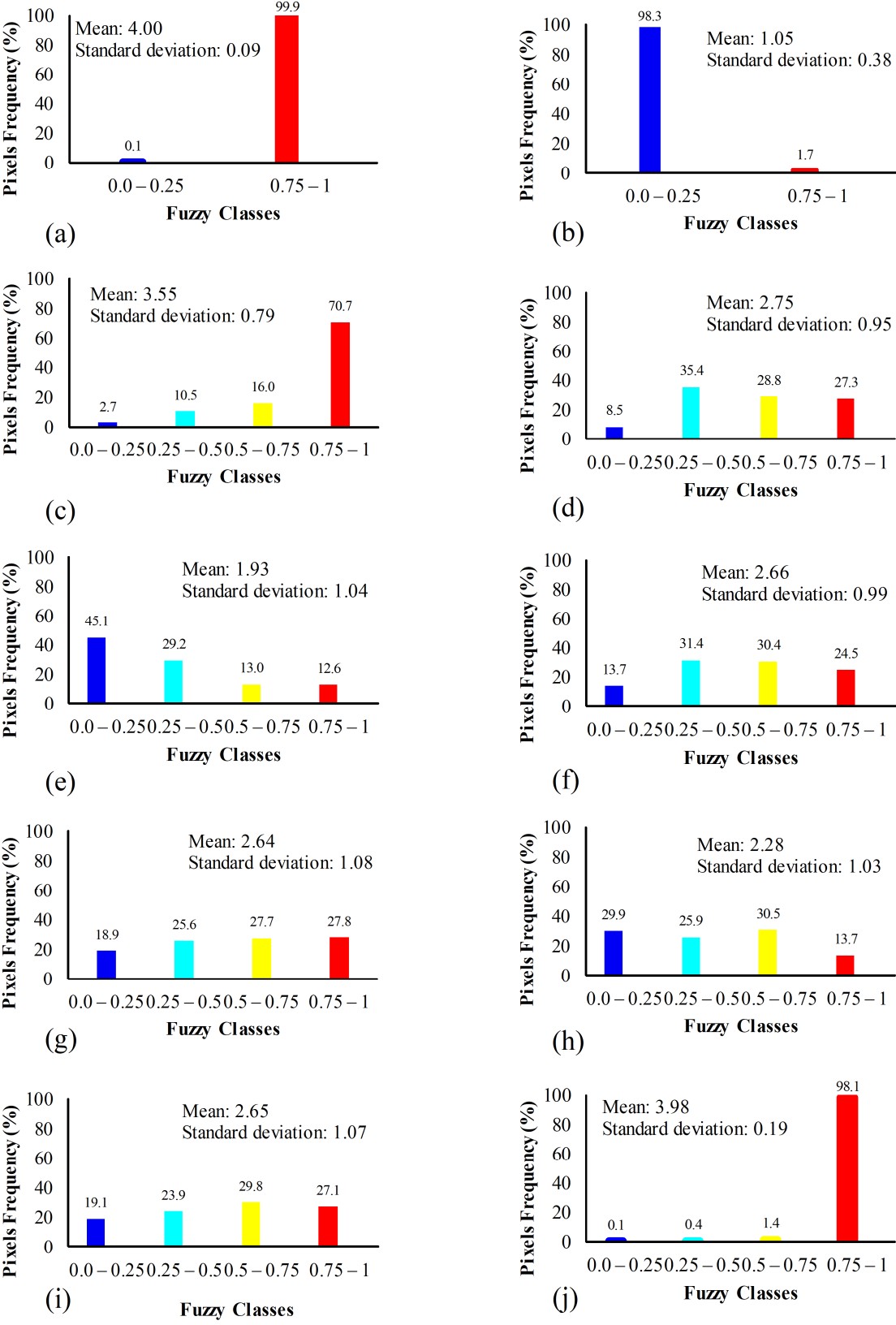

**Figure 5.** Pixels frequency for each fuzzy class. (**a**) Compactness coefficient; (**b**) slope; (**c**) Digital Elevation Model (DEM); (**d**) annual average maximum rainfall; (**e**) dry period average maximum rainfall; (**f**) rainy period average maximum rainfall; (**g**) annual accumulated rainfall; (**h**) dry period accumulated rainfall; (**i**) rainy period accumulated rainfall; (**j**) proximity to hydrographies.

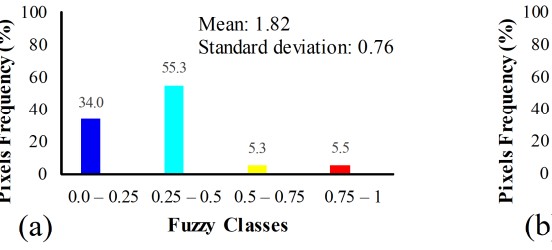
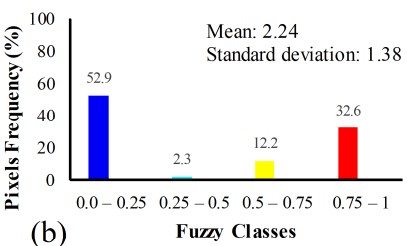
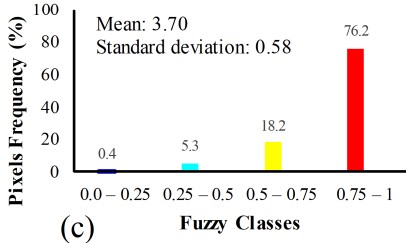

**Figure 6.** Pixels frequency for each fuzzy class. (**a**) Land use and occupation; (**b**) soil classes; (**c**) proximity to water masses.

*Flood Vulnerability in Espírito Santo State*

In Espírito Santo state, some sites are more susceptible to the flooding negative impacts than others are. Flooding occurs mainly in areas where the population is located near watercourses and in less sloping places. The flood vulnerability spatialization for the study area regarding the dry season is presented in Figure 7.

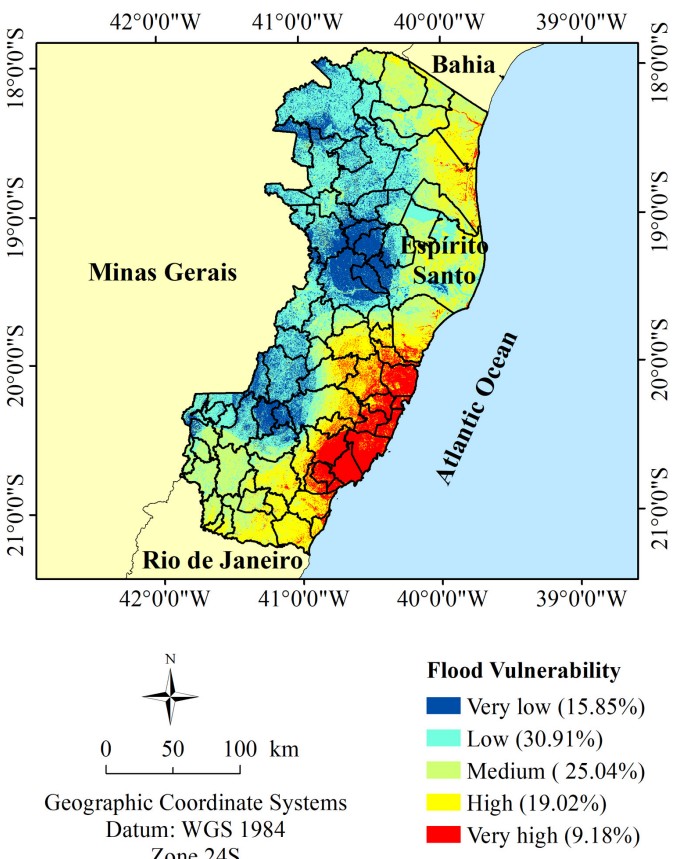

**Figure 7.** Dry period flood vulnerability for Espírito Santo state.

Espírito Santo state has a large area corresponding to a low flood vulnerability class, with a value of 30.91%, which is followed by medium flood vulnerability class, with a value of 25.04%. Most of these areas correspond to the state's western region.

The areas that were classified as having a very high flood vulnerability represent 9.18% and they are located in coastal and metropolitan state regions, including the municipalities of Serra (483.85 km$^2$), Vitória (79.87 km$^2$), Vila Velha (195.70 km$^2$), Guarapari (565.46 km$^2$), Anchieta (403.62 km$^2$), Piúma (71.23 km$^2$) and Iconha (184.14 km$^2$) (Table S1, Supplementary Materials), which are classified as very high flood vulnerability class practically throughout the entire extension of the municipalities.

For the rainy season (Figure 8), the medium flood vulnerability class presents the largest area, corresponding to 33.73%. The greater rainfall presence provides differences in relation to the dry season flood vulnerability classification. For instance, the municipalities in the northern region that were previously classified as having a low flood vulnerability are now classified as having a medium flood vulnerability during the rainy season, particularly the Ecoporanga, Montanha and Mucurici municipalities.

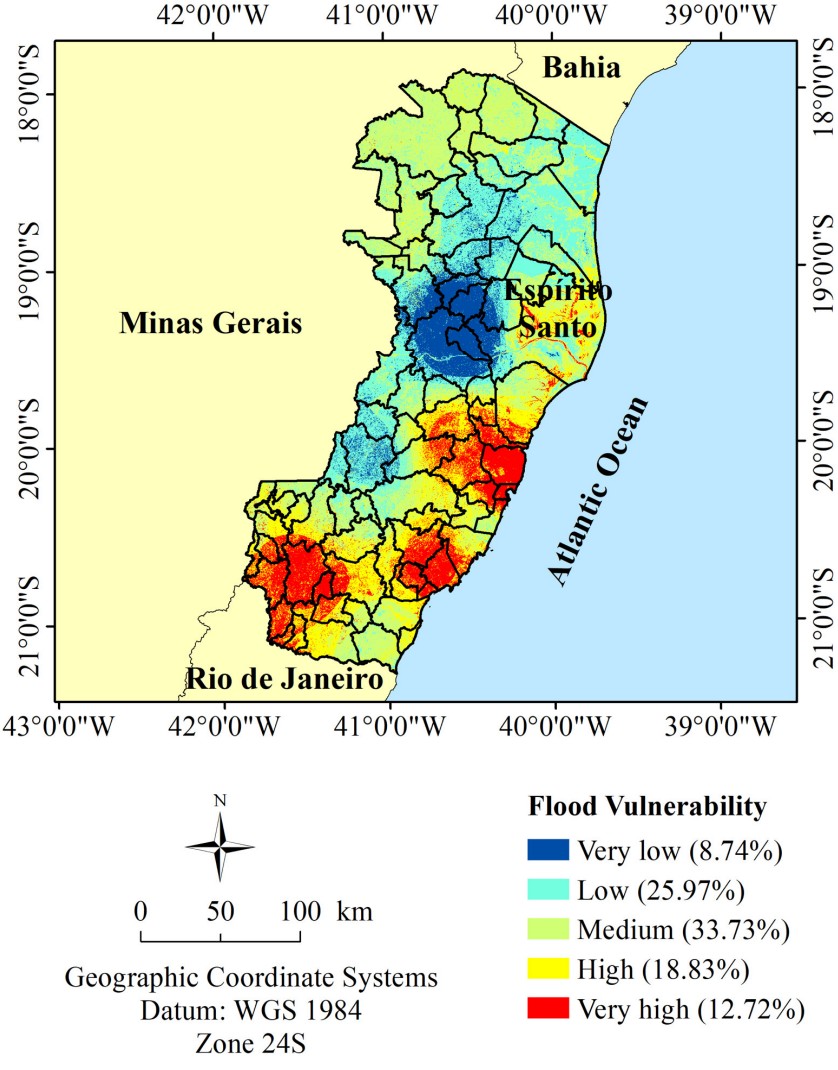

**Figure 8.** Rainy period flood vulnerability for Espírito Santo state.

The municipalities of Alegre, Guaçuí, Jerônimo Monteiro, São José do Calçado, Bom Jesus do Norte, Dores do Rio Preto, Ibitirama, Muniz Freire, Divino de São Lourenço, Cachoeiro de Itapemirim, Apiacá and Muqui are classified as having a very high flood vulnerability. Among these municipalities, Guaçuí (342.28 km$^2$), Alegre (685.34 km$^2$), Jerônimo Monteiro (151.07 km$^2$), São José do Calçado (198.78 km$^2$) and Bom Jesus do

Norte (51.66 km²) stand out since they also present the same flood vulnerability class in approximately all of their territorial extensions.

Similarly, some coastal and metropolitan municipalities that were already in the very high flood vulnerability class the during dry season remain with the same classification for the rainy season practically throughout their entire territorial extension, such as Serra (514.97 km²), Vitória (84.00 km²), Anchieta (359.81 km²), Iconha (163.41 km²) and Piúma (61.13 km²) (Table S1, Supplementary Materials). These very high flood vulnerability regions coincide with the areas presenting lower altitudes and lower slopes, which provide a greater susceptibility to flooding.

The annual flood vulnerability, shown in Figure 9, is similar to the dry season vulnerability. Approximately 29.15% of the areas are classified as having a low flood vulnerability, which is followed by 28.06% of them having a medium flood vulnerability and 25.98% of them having high flood vulnerability. In general, the medium flood vulnerability category includes most of the municipalities in state southern region, while the very high flood vulnerability category includes the municipalities in the coastal and metropolitan state regions.

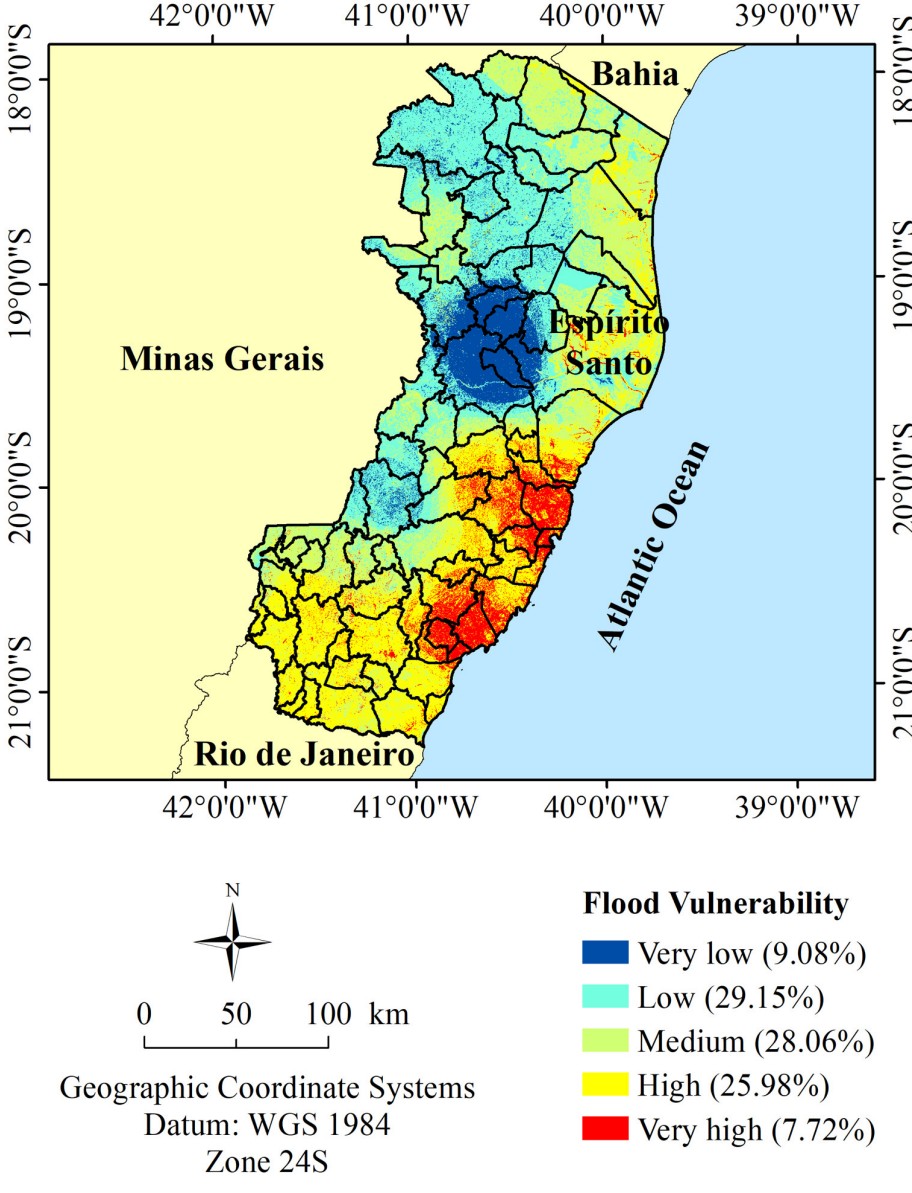

**Figure 9.** Annual flood vulnerability for Espírito Santo state.

Among the municipalities that are classified as very high flood vulnerability, there are Serra (444.74 km$^2$), Vitória (70.75 km$^2$), Anchieta (342.46 km$^2$), Iconha (167.23 km$^2$) and Piúma (68.97 km$^2$), with approximately the entire extension of their territories being classified as this, in addition to a considerable part of the Santa Leopoldina (333.98 km$^2$) and Alfredo Chaves (301.38 km$^2$) municipalities (Table S1, Supplementary Materials).

The very low vulnerability class during the rainy period (8.74%) encompasses the same municipalities as they do during dry and rainy periods, which are São Domingos do Norte (278.31 km$^2$), Governador Lindemberg (344.77 km$^2$), Marilândia (297.30 km$^2$) and Colatina (1081.81 km$^2$) (Table S1, Supplementary Materials).

## 4. Discussion

Based on the obtained results, it was observed that areas with the greater degree of flood vulnerability during the dry season are concentrated in the state metropolitan and coastal regions. The results differ for the rainy season, with the places that were previously considered as low risk migrating to the high to very high risk categories, such as in the southern region. In the annual period, the high and very high risks are distributed throughout the state's central and southern regions, and together, they account for 33.7% of the flood vulnerability, while northwestern region presents a low risk.

It is possible to observe that in the three assessed scenarios (the dry, rainy and annual periods), the metropolitan regions always fall into the very high risk class. This result may be linked to significant land use and occupation changes in these regions, with growing urban expansions, that are often disorderly, which consequently hinder the water infiltration process. The authors of [32] believe that the advances in urbanization, especially in improper areas, increase the disaster danger and risk.

According to [5], other factors may influence the flood vulnerability in urban areas, such as the continued densification of residential areas, infrastructure development and urban sprawl. The authors of [33] also state that socioeconomic growth is a dominant factor for flood vulnerability increase.

The results of this work corroborate those that were found by the authors of [34], who evaluated the flood risk in China. According to the authors, the areas that are at greatest risk are those that have the highest amount of rainfall and are distributed on low, flat terrain and are close to rivers and the coast.

In general, urban areas are the ones that suffer the most from flood occurrence due to a combination of rapid urban growth, heavy rainfall, overflowing dams and construction on floodplains or in the lower areas of cities [35]. According to [15], rapid urbanization causes an increase in the flood risk due to drastic changes in the land use/cover in these areas [36].

Another factor that may be associated with and may contribute to the increase in the flood vulnerability is global climate changes, which are caused mainly by meteorological phenomena arising from anthropogenic impacts on the globe, in addition to the presence of climatic events such as El Nino and Lã Nina, that cause a series of changes that are related to temperature and rainfall, therefore, directly influencing floods occurrence. The authors of [2] corroborate that climate changes have a significant impact on flood vulnerability due to the changes in weather patterns, such as rising sea levels, which make areas that are close to the coast even more vulnerable to flooding phenomena.

However, the regions that are further away from the coast also draw attention to the need of mitigating these impacts to reduce the damage and risk of possible flooding phenomena. As each municipality has been assessed individually, thereby obtaining the areas of flood vulnerability for the three studied periods, the municipalities of Alegre and Cachoeiro de Itapemirim stand out, both with significant riparian occupations and with high rates of flooding-vulnerable areas during the rainy periods. The relief factor may be exerting greater influence in relation to flood vulnerability, coinciding with expressive areas that present floodable perimeters, such as the coastal and metropolitan regions, which present a greater degree of flood vulnerability and that also are located at low altitudes and low slopes.

São Domingos do Norte, Governador Lindemberg, Marilândia and Colatina municipalities are classified as having a very low risk during the dry, rainy and annual periods. These municipalities presented lower values for accumulated rainfall during the dry, rainy and annual periods and they possess the Oxisol soil class. Oxisols are well-weathered soils that are associated with good permeability and present low moisture retention [37]. These soil physical properties, specifically the good soil infiltration capacity combined with the low annual accumulated rainfall, may explain them having very low flood vulnerability for the three assessed periods.

Flooding is a catastrophic event that can occur anywhere in the world. In addition, these catastrophes are the result of a combination of natural factors and human actions, such as infrastructure development and urban sprawl [5,36]. In this sense, prediction models are important strategies for knowing the areas that are the most susceptible to flooding [34,38] and to assist in public management decision making that is aimed at preventing and mitigating the risks and consequences of floods for the populations that are living in the areas that are considered to be at risk.

## 5. Conclusions

Considering the three scenarios, it is concluded that Espírito Santo state presents, in most of its area, as having a low (29.15%) and medium (28.06%) flood vulnerability. However, both the metropolitan region (Vitória, Vila Velha, Viana, Serra and Cariacica) and part of the coast and immigration region (Piúma, Anchieta and Iconha) represent almost the entire area that was classified as very high flood vulnerability throughout the year (7.72%).

This work results demonstrate the importance of applying geotechnologies on flood vulnerability zoning. In this context, it is concluded that vulnerability mapping is a fundamental auxiliary tool in decision making by government agencies, aiming to minimize the risks and consequences of floods. Zoning also allows the public policy creation for better urban planning and investment in environmental education. In addition, the used methodology has potential for its application and adaptation to other areas of study around the world.

**Supplementary Materials:** The following supporting information can be downloaded at: https://www.mdpi.com/article/10.3390/w14223731/s1, Figure S1: Rainfall distribution for Espírito Santo state, Brazil. (a) Annual average maximum rainfall, (b) dry period average maximum rainfall, (c) rainy period average maximum rainfall, (d) annual accumulated rainfall, (e) dry period accumulated rainfall (f) rainy period accumulated rainfall; Figure S2: Digital Elevation Model for Espírito Santo state, Brazil; Figure S3: Slope classes for Espírito Santo state, Brazil, as proposed by EMBRAPA, (1979); Figure S4: Soil types for Espírito Santo state, Brasil; Figure S5: Land use and occupation for Espírito Santo state, Brazil; Figure S6: Demographic density (people/km$^2$) and municipalities identification for Espírito Santo state, Brazil; Table S1: Flood vulnerability area, in square kilometers, by municipality for Espírito Santo state, Brazil. Ref [39] is citied in Supplementary Materials.

**Author Contributions:** Conceptualization, R.d.C.F.C., T.R.M., K.B.d.S. and A.R.d.S. (Alexandre Rosa dos Santos); methodology, R.d.C.F.C., T.R.M., K.B.d.S. and A.R.d.S. (Alexandre Rosa dos Santos); software, G.A.C., S.S.Z., K.V.B., C.B.C.F. and M.R.M.; validation, G.A.C., S.S.Z., K.V.B., C.B.C.F. and M.R.M.; formal analysis, R.d.C.F.C., T.R.M., K.B.d.S., G.A.C., S.S.Z., K.V.B., C.B.C.F., M.R.M. and A.R.d.S. (Alexandre Rosa dos Santos); investigation, A.R.d.S. (Aline Ramalho dos Santos), A.H.C.R., E.S.A.F., E.F.A. and F.P.d.N.; resources, J.F.V.d.L., J.S.M. and L.L.B.; data curation, M.O.A., R.G.G. and S.M.d.S.G.; writing—original draft preparation, R.d.C.F.C., T.R.M., K.B.d.S., G.A.C., S.S.Z., K.V.B., C.B.C.F., M.R.M. and A.R.d.S. (Alexandre Rosa dos Santos); writing—review and editing, R.d.C.F.C., T.R.M. and P.A.G.F.; visualization, R.d.C.F.C., T.R.M. and P.A.G.F.; supervision, A.R.d.S. (Alexandre Rosa dos Santos); project administration, A.R.d.S. (Alexandre Rosa dos Santos); funding acquisition, A.R.d.S. (Alexandre Rosa dos Santos). All authors have read and agreed to the published version of the manuscript.

**Funding:** This research received no external funding.

**Data Availability Statement:** Not applicable.

**Acknowledgments:** The authors thank the following agencies for making the database available for the research development: (a) Geospatial Bases Integrated System of Espírito Santo State (GEOBASES), (b) Brazilian Institute of Geography and Statistics (IBGE), (c) Institute of Agricultural and Forest Defense of Espírito Santo State (IDAF), (d) Capixaba Institute of Research, Technical Assistance and Rural Extension (INCAPER) and (e) Meteorology National Institute (INMET). The authors also thank the following research and development agencies for their assistance, funding and support in the research development: (a) Higher Education Personnel Improvement Coordination (CAPES), (b) Research and Innovation Support Foundation of Espírito Santo (FAPES) and (c) National Council for Scientific and Technological Development (CNPq). The present work was carried out with the support of the Fundação de Amparo à Pesquisa e Inovação do Espírito Santo (FAPES)—FAPES Public Notice No. 07/2022—Publication of Technical-Scientific Articles. Finally, the authors thank the Postgraduate Program in Forest Science of Espírito Santo Federal University and the CNPq registered research group Geotechnology Applied to Global Environment (GAGEN).

**Conflicts of Interest:** The authors declare no conflict of interest.

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
