# Peer review of "GIS-Based Approach Applied to Study of Seasonal Rainfall Influence over Flood Vulnerability"

_water, doi:10.3390/w14223731_

Round 1

Reviewer 1 Report

The manuscript "GIS-based approach applied to study of seasonal rainfalls influence over flood vulnerability" is well written and the methodology is timely. The objective was fulfilled. However, it needs several improvements if it is considered for publication.

1. Enhance the summary with statistical data.
2. More current bibliographical references are needed.
3. Revision of the English in the manuscript.
4. Maybe write the name of the study area: Espírito Santo on the figure 5.
6.What were the satellite images and spatial resolution of the Land Use.
7. The web pages of the data sources should be presented.
8. Greater explanation in obtaining hydrography. Why was the Euclidean distance applied?

Reviewer 2 Report

No comments

Reviewer 3 Report

The article needs a very profound revision of the english.

1.Introdution

The introduction is short and the reading list associated is not  easily available for the reviewers.

2 Materials and methods

Study area is too short. A brief characterisation of the area will be  needed.

The methodology seems adequate

The presentation of results are in accordance with the methods.

 –The Discussion  could be improved  in establishing  relationships with other papers in tropical areas.

 Conclusions are fine.

Round 2

Reviewer 1 Report

Dear authors, thank you for considering my suggestions.

Reviewer 3 Report

The revision by the authors  mets the recomendations suggested.